# Antiviral Activity of Porcine IFN-λ3 and IFN-α against Porcine Rotavirus In Vitro

**DOI:** 10.3390/molecules27144575

**Published:** 2022-07-18

**Authors:** Lishuang Deng, Yue Yin, Zhiwen Xu, Fengqin Li, Jun Zhao, Huidan Deng, Zhijie Jian, Siyuan Lai, Xiangang Sun, Ling Zhu

**Affiliations:** 1College of Veterinary Medicine, Sichuan Agricultural University, Chengdu 625014, China; 18428374864@163.com (L.D.); daiyi_0@163.com (Y.Y.); abtcxzw@126.com (Z.X.); zhaojunjoy@126.com (J.Z.); denghuidan@sicau.edu.cn (H.D.); jianzhijie134@163.com (Z.J.); sicaulaisiyuan@163.com (S.L.); sun.xian.gang@163.com (X.S.); 2Key Laboratory of Animal Disease and Human Health of Sichuan Province, Sichuan Agricultural University, Chengdu 625014, China; 3College of Animal Science, Xichang University, Xichang 615000, China; lfqsean@126.com

**Keywords:** antiviral activity, IFN-λ3, IFN-α, PoRV

## Abstract

Interferons (IFNs) play a major role in the host’s antiviral innate immunity. In response to viral infection, IFNs bind their receptors and initiate a signaling cascade, leading to the accurate transcriptional regulation of hundreds of IFN-stimulated genes (ISGs). Porcine rotavirus (PoRV) belongs to genus *Rotavirus* of the *Reoviridae* family; the infection is a global epidemic disease and a major threat to the pig industry. In this study, we found that IFN-λ3 inhibited the replication of PoRV in both MA104 cells and IPEC-J2 cells, and this inhibition was dose-dependent. Furthermore, the antiviral activity of IFN-λ3 was more potent in IPEC-J2 cells than in MA104 cells. Further research showed that IFN-λ3 and IFN-α might inhibit PoRV infection by activating ISGs, i.e., MxA, OASL and ISG15, in IPEC-J2 cells. However, the co-treatment of IFN-λ3 and IFN-α did not enhance the antiviral activity. Our data demonstrated that IFN-λ3 had antiviral activity against PoRV and may serve as a useful antiviral candidate against PoRV, as well as other viruses in swine.

The host innate immune system is the first line of defense against viral infections. Interferons (IFNs) play a major role in the host’s antiviral innate immunity and are broadly classified into three distinct types based on their molecular architecture, pathway induction, and cell-receptor specificity [1,2]. At present, type I IFN (IFN-α, IFNβ, IFN-ε, IFN-κ and IFN-ω); type II IFN (IFN-γ); and type III IFN (IFN-λ) have been identified [3,4]. Type III IFNs were recently discovered as a unique class of antiviral factors, which consists of IFN-λ1, IFNλ2, IFN-λ3 and IFN-λ4 in humans; IFN-λ2 and IFN-λ3 in mice; and IFN-λ1, IFN-λ3 and IFN-λ4 in swine [3,5,6]. In response to viral infection, IFNs that are produced by the host bind their receptors and initiate a signaling cascade, leading to the accurate transcriptional regulation of hundreds of IFN-stimulated genes (ISGs) [7]. Classical ISGs belong to three gene families: Mx proteins; 2′,5′-oligoadenylate synthetase; or ds RNA-activated protein kinase [8]. Notably, type I and III IFNs differ greatly in sequence and structure, as well as using different receptors for their signaling, but the downstream signaling pathways and transcriptional responses that are activated by them are remarkably similar [9]. The ISGs that are induced by type III IFNs are a subset of those that are induced by type I IFNs [10]. However, type I and type III IFNs show differences in terms of both regulation and biological activity, which may be related to differences in the magnitude and kinetics of signaling and in the types of cells that respond to type I versus type III IFNs [10]. In general, type III IFNs have slower kinetics and a lower amplitude of individual ISGs expression, compared to type I IFNs [11]. However, there are exceptions; for example, the expression of IFN-α and IFN-λ in plasmacytoid dendritic cells after infection with influenza A virus in vitro followed highly similar kinetics [12]. Recent studies have demonstrated that type I and type III IFNs have tissue- and cell-specific differences due to their respective receptor distribution. The antiviral effects of type III IFNs rather than type I IFNs are particularly evident at epithelial barriers, such as the gastrointestinal, respiratory and reproductive tracts [10].

Porcine rotavirus (PoRV) belongs to genus *Rotavirus* of family *Reoviridae*, which infects pigs of all ages—especially piglets—and causes diarrhea, vomiting and dehydration [13]. Since PoRV was first isolated from swine waste in 1974, PoRV infection has become a global epidemic disease and is currently a major threat to the pig industry [14]. PoRV primarily invades the mature enterocytes that line the villi of the jejunum and ileum [15]. The intestinal porcine epithelial cell line J2 (IPEC-J2) derives from the jejunum of newborn pigs, which preserves most of the characteristics of mature enterocytes and is an appropriate model for studying the interaction of intestinal immune responses and host-pathogens in vitro [16,17]. In this study, the antiviral activity of IFN-λ3 against PoRV in MA104 cells and IPEC-J2 cells was evaluated. The possible role of IFN-α and IFN-λ3 in the replication of PoRV and the expressions of ISGs genes that are induced by IFNs were also investigated.

In the current study, PoRV SC-R strain was used as a model virus to evaluate the antiviral activity of IFN-α and IFN-λ3. The virus was propagated and titrated in MA104 cells. Recombinant porcine IFN-α and IFN-λ3 were expressed in *E. coli* and stored in our laboratory [18]. To explore the anti-PoRV effect of different concentrations of IFN-λ3, MA104 cells and IPEC-J2 cells were untreated or pre-treated with IFN-λ3 (10, 100, 1000 ng/mL) for 24 h. Then, the cells were infected with PoRV SC-R strain at an MOI of 0.1 for 24 h. The cytopathic effect (CPE) units in culture plates were counted. To further determine the antiviral effect of IFN-λ3, the IPEC-J2 cells were pre-treated with 100 ng/mL of IFN-λ3 for 24 h, and then infected with PoRV SC-R strain for 12, 24, 36 h at a MOI of 0.1. The quantification of PoRV VP6 mRNA was performed on total cellular supernatant RNA. The primers of the VP6 gene are listed in Table 1. As shown in Figure 1a, IFN-λ3 inhibited the replication of PoRV in a dose-dependent manner in MA104 cells and IPEC-J2 cells. IFN-λ3 exhibited a more potent activity against PoRV infection in IPEC-J2 cells compared with that in MA104 cells. Importantly, IFN-λ3 significantly reduced the copies of PoRV infection in IPEC-J2 cells at different time points (Figure 1b).

As major members of the IFN family, IFN-α and IFN-λ3 play important roles in innate immunity against various viral infections in pigs [19,20]. Therefore, we wondered whether a co-treatment of IFN-λ3 and IFN-α could enhance the antiviral efficacy against PoRV in IPEC-J2 cells. The IPEC-J2 cells were treated with different concentrations of recombinant IFN-α (10, 100 and 1000 IU/mL); IFN-λ3 (0.1, 1 and 10 ng/mL); and mixtures of IFN-α + IFN-λ3 (10 IU/mL + 0.1 ng/mL, 100 IU/mL + 1 ng/mL, 1000 IU/mL + 10 ng/mL) for 24 h, respectively. Then, the cells were infected with PoRV SC-R strain for 36 h at an MOI of 0.01. A low MOI could obtain a higher resolution of the virus yield inhibition curves. Our previous study showed that after inoculation with 0.01 and 0.1 MOI of PoRV SC-R strain, the virus titers increased in the first 36 h. The titer of virus with an MOI of 0.01 was higher than that of virus with an MOI of 0.1 at 36 h (data not shown). The virus titer in the supernatant was titrated by TCID_50_. As shown in Figure 2, IFN-λ3 and IFN-α could both inhibit the replication of PoRV in IPEC-J2 cells in a dose-dependent manner. However, the co-treatment of IFN-λ3 and IFN-α did not enhance the antiviral efficacy.

To further explore the possible mechanisms of IFN-α and IFN-λ3 anti-PoRV infection, we examined the expression of ISGs that were induced by IFNs. The IPEC-J2 cells were incubated with different IFN doses (either individually or in combination) for 24 h. The total cellular RNA was extracted and the ratio of mRNA for ISG15, MxA and OASL to β-actin was calculated. These ISGs primers are listed in Table 1. As shown in Figure 3, a dose-dependent induction of mRNA for ISG15, MxA and OASL was observed in the IPEC-J2 cells that were treated with both the single IFN and the co-treatment of IFN-λ3 and IFN-α. These results indicate that IFN-λ3 and IFN-α might inhibit PoRV infection by activating ISGs in IPEC-J2 cells. Unfortunately, the potential mechanisms of antagonism between IFN-α and IFN-λ3 are still not clear.

In summary, our data demonstrated that IFN-λ3 had the ability to inhibit the replication of PoRV in both MA104 cells and IPEC-J2 cells, and this inhibition was dose-dependent. Furthermore, the antiviral activity of IFN-λ3 was more potent in IPEC-J2 cells than in MA104 cells. We also found that IFN-λ3 and IFN-α might inhibit PoRV infection by activating ISGs, i.e., MxA, OASL and ISG15, in IPEC-J2 cells. However, the co-treatment of IFN-λ3 and IFN-α did not enhance the antiviral activity. The exact mechanism is yet to be elucidated. We deduced that the kinetics differences of ISGs that are induced by different types of IFNs may play a significant role. Our data demonstrated that IFN-λ3 had antiviral activity against PoRV and may serve as a useful antiviral candidate against PoRV, as well as other viruses in swine.

## Figures and Tables

**Figure 1 molecules-27-04575-f001:**
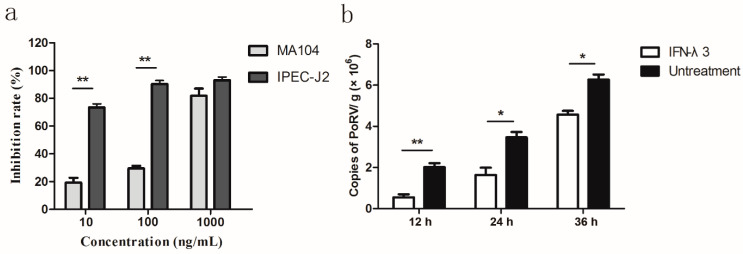
Porcine IFN-λ3 inhibited PoRV infection in MA104 cells and IPEC-J2 cells. (**a**) MA104 cells and IPEC-J2 cells were stimulated with porcine IFN-λ3 at different concentrations for 24 h and then infected with PoRV at 0.1 MOI for 36 h. (**b**) IPEC-J2 cells were treated or untreated with 100 ng/mL of IFN-λ3 for 24 h and then infected with PoRV at 0.1 MOI for 12, 24 or 36 h, respectively. PoRV VP6 mRNA was detected by an RT-PCR. Data were presented as mean ± SEM (*n* = 3). * *p* < 0.05; ** *p* < 0.01 by t test.

**Figure 2 molecules-27-04575-f002:**
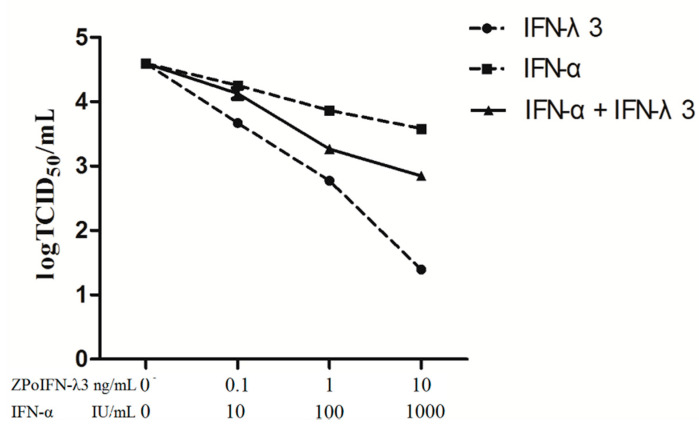
Inhibition of PoRV replication by porcine IFN-α, IFN-λ3 and IFN-α + IFN-λ3. IPEC-J2 cells were stimulated with either one IFN alone or in combination with two IFNs for 24 h, and then infected with PoRV at 0.01 MOI for 36 h. The PoRV titer in the supernatant was titrated by TCID_50_. Dotted lines: IFN-α (■) or IFN-λ3 (●) used individually; continuous line: IFN-α and IFN-λ3 (▲) used in combination. Data were presented as mean ± SEM (*n* = 3).

**Figure 3 molecules-27-04575-f003:**
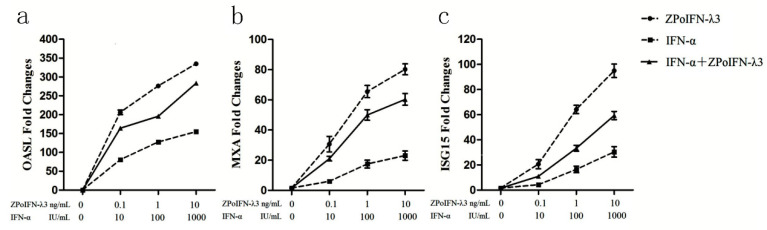
Expressions of ISGs induced by IFN-α, IFN-λ3 and IFN-α + IFN-λ3 in IPEC-J2 cells. IPEC-J2 cells were stimulated with either one IFN alone or in combination with two IFNs for 24 h, and the mRNA levels of OASL (**a**), MxA (**b**) and ISG15 (**c**) were measured by a relative RT-qPCR. The results were normalized by the β-actin levels of each sample. Dotted lines: IFN-α (■) or IFN-λ3 (●) used individually; continuous line: IFN-α and IFN-λ3 (▲) used in combination. Data were presented as mean ± SEM (*n* = 3).

**Table 1 molecules-27-04575-t001:** Primers used in this study.

Gene Name	Primer Name	Sequence (5′–3′)	Product Size (bp)
VP6	VP6-F	TTCGGATTACTTGGCACTA	118
	VP6-R	TAGCCATTTCATCCATACAC	
ISG15	ISG15-F	ACAAGGGTCGCAGCAACGC	192
	ISG15-R	GCAGATTCATATACACGGTG	
MxA	MxA-F	GATGAAAGCGGGAAGATG	119
	MxA-R	TTGGTAAACAGCCGACAC	
OASL	OASL-F	TCCTTCGCCAAGTTACAG	136
	OASL-R	CATAGAGAGGGGGCAGCC	
β-actin	β-actin-F	ATCGTGCGGGACATCAAG	179
	β-actin-R	GGAAGGAGGGCTGGAA	

## Data Availability

The data used to support the findings of this study are available. Further inquiries can be directed to the corresponding authors.

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
