# Peer review of "Antiviral Activity of Porcine IFN-λ3 and IFN-α against Porcine Rotavirus In Vitro"

_molecules, 2022, doi:10.3390/molecules27144575_

Round 1

Reviewer 1 Report

This work is devoted to investigating the antiviral activity of porcine IFN-λ3 and IFN-α against porcine rotavirus by in vitro study.

The subject of the manuscript addressed by the authors is interesting and relatively comprehensive presented.

I would highlight the following points that it is desirable to improve and increase the interest for the paper:

-a time of addition experiments should be presented;

-the use of a lower MOI in the experiment presented in Figure 2 should be better argued.

Based on my comments the paper needs the minor improvements mentioned above before publication

Author Response

Dear Reviewer:

Thank you for your comments for our manuscript entitled “Antiviral activity of porcine IFN-λ3 and IFN-α against porcine rotavirus in vitro” (Submission ID: molecules-1794315). And we would like to thank you for your careful reading, helpful comments, and constructive suggestions, which has significantly improved the presentation of our manuscript. We have carefully considered all comments from the reviewers, and revised our manuscript accordingly and marked in red. We believe that our responses have well addressed all concerns from the reviewers. All corrections in the paper and the responds to the reviewer's comments are as following:

Reviewer 1

This work is devoted to investigating the antiviral activity of porcine IFN-λ3 and IFN-α against porcine rotavirus by in vitro study. The subject of the manuscript addressed by the authors is interesting and relatively comprehensive presented. I would highlight the following points that it is desirable to improve and increase the interest for the paper:

  1. A time of addition experiments should be presented;

Response: Thank you very much for your reminding. We have detailed the time of IFNs pretreatment and the time of virus infection in the text and Figures' legends in the revised manuscript.

  1. The use of a lower MOI in the experiment presented in Figure 2 should be better argued.

Response: Thank you for suggestion. We have supplemented the reason for using 0.01 MOI in the experiment presented in Figure 2 in the revised manuscript. Our previous study showed that after inoculation with 0.01 and 0.1 MOI of PoRV SC-R strain, the virus titers were on the rise in the first 36 h. And the titer of virus with MOI of 0.01 was higher than that of virus with MOI of 0.1 at 36 h. Therefore, we chose 0.01 MOI as the infectious dose of the virus.

Reviewer 2 Report

In this manuscript, authors demonstrated IFN-λ3 inhibited the replication of PoRV both in MA104 cells and IPEC-J2 cells in a dose-dependent manner. The antiviral activity of IFN-λ3 was more potent in IPEC-J2 cells than that in MA104 cells. Further, authors showed that IFN-λ3 and IFN-α inhibits PoRV infection by activating ISGs, i.e. MxA, OASL and ISG15, in IPEC-J2 cells. However, the co-treatment of IFN-λ3 and IFN-α didn’t enhance the antiviral activity. These data demonstrated that IFN-λ3 had antiviral activity against PoRV, suggest IFN-λ3 may be a useful antiviral agent against PoRV in swine. I recommend authors add a review on whether differences in type I and type III function, signaling potency and kinetics or tissue-specific responsiveness in the introduction section, and needs to mention whether any specific ISGs are uniquely induced by type I versus type III IFNs, allows readers to understanding what's the significance and scientific value of this study.  suggesting add the reference Helen M.Lazear14John W.Schoggins2Michael S.Diamond3, Shared and Distinct Functions of Type I and Type III Interferons Immunity Volume 50, Issue 4, 16 April 2019, Pages 907-923 https://doi.org/10.1016/j.immuni.2019.03.025

Author Response

Dear Reviewer,

Thank you for your comments for our manuscript entitled “Antiviral activity of porcine IFN-λ3 and IFN-α against porcine rotavirus in vitro” (Submission ID: molecules-1794315). And we would like to thank you for your careful reading, helpful comments, and constructive suggestions, which has significantly improved the presentation of our manuscript. We have carefully considered all comments from the reviewers, and revised our manuscript accordingly and marked in red. We believe that our responses have well addressed all concerns from the reviewers. All corrections in the paper and the responds to the reviewer's comments are as following:

Reviewer 2

In this manuscript, authors demonstrated IFN-λ3 inhibited the replication of PoRV both in MA104 cells and IPEC-J2 cells in a dose-dependent manner. The antiviral activity of IFN-λ3 was more potent in IPEC-J2 cells than that in MA104 cells. Further, authors showed that IFN-λ3 and IFN-α inhibits PoRV infection by activating ISGs, i.e. MxA, OASL and ISG15, in IPEC-J2 cells. However, the co-treatment of IFN-λ3 and IFN-α didn’t enhance the antiviral activity. These data demonstrated that IFN-λ3 had antiviral activity against PoRV, suggest IFN-λ3 may be a useful antiviral agent against PoRV in swine. I recommend authors add a review on whether differences in type I and type III function, signaling potency and kinetics or tissue-specific responsiveness in the introduction section, and need to mention whether any specific ISGs are uniquely induced by type I versus type III IFNs, allowing readers to understand what's the significance and scientific value of this study. Suggesting add the reference Helen M. Lazear, John W. Schoggins and Michael S. Diamond. Shared and Distinct Functions of Type I and Type III Interferons. Immunity, Volume 50, Issue 4, 16 April 2019, Pages 907-923. https://doi.org/10.1016/j.immuni.2019.03.025.

Response: Thank you very much for your suggestion. We are sorry for our neglect. We have outlined differences in type I and type III function, signaling potency, kinetics, and tissue-specific responses in the Introduction section of the revised manuscript, and marked them in red.

Round 2

Reviewer 2 Report

The revised manuscript has been improved, and is acceptable for publication.